# Health as Battlefield: News and Misinformation in the Early Stage of COVID-19 Outbreak

**DOI:** 10.3390/ijerph19169800

**Published:** 2022-08-09

**Authors:** Qian Liu, Fan Yang

**Affiliations:** 1School of Journalism and Communication, National Media Experimental Teaching Demonstration Center, Jinan University, Guangzhou 511433, China; 2Department of Communication, University at Albany, State University of New York, Albany, NY 12222, USA

**Keywords:** coronavirus, COVID-19, LDA topic modeling, misinformation, news, sentiment analysis

## Abstract

From the epidemic center in Wuhan to the entirety of China, with the growing infected population, people are seeking and processing health-related information both online and from traditional media outlets such as newspapers. Online misinformation regarding COVID-19 has been influencing a wide range of readers demonstrating general citizens’ virus-related concerns, while press media have been actively participating in health communication in an attempt to build up a robust, harmonious, and healthy environment. Via a comparison between the news data with the misinformation data during the early stage of the COVID-19 pandemic, from 1 January 2020 to 20 February 2020, we conducted an LDA topic-modeling analysis and a sentiment analysis. This study sheds light on the nature of people’s methods of health communication with online and press media sources during the early period of the pandemic crisis and provides possible readable explanations for the driving force of misinformation and the emotional changes experienced by the public.

## 1. Introduction

Misinformation during crises form and develop on viral platforms for many reasons. Some scholars have studied misinformation and rumors in crises during wartime [1], others have focused on the pandemic, natural disasters [2], or terrorist attacks [3]. Scholars found that online sites and mainstream newspapers circulated several pieces of misleading information during the 9–11 crisis [3]. Social media also serves as an arena for the spreading of rumors, such as twitter [4,5,6], where social media accelerates the communication process of crisis responses and becomes a part of it [4]. It has also been documented that the dissemination of rumors on twitter platforms are different from news in the same community; for example, people questioned misinformation more on social media compared to news outlets [5].

The development and spread of misinformation on the internet places public interests at risk and diminishes institutional authority. A lack of detailed information assists the spread of misinformation and causes a great deal of trouble with respect to public health communication. According to scholars, web-based news information in crises have a mixed and contradictory reputation, where some online news sources lack details of the crisis, whereas others lack relevant information in various formats, besides textual information, such as visual, audio, and video [2]. A lack of detailed information assists the spreading of uncertain information, rumors, and misinformation. Lies spread at a faster speed than truth [7].

## 2. Related Works

For decades, researchers have been looking for features of the spread of misinformation online, so that the best practices could be adopted to fight both viruses and misinformation. Some scholars have analyzed social media messages for their rumor-affirming or rumor-correcting speeds to establish a spreading model of fake news online [7]. Others have discovered that the factors causing social media-based misinformation are information without a clear source, stories with a great deal of personal involvement, and anxiety [8]. The reason for this comparison of the traditional news with the misinformation online is that due to the structural differences between the two (e.g., readership, ownership, etc.), traditional news media can only cover certain issues related to the epidemic but not others (e.g., the lab-leaking theory) while online misinformation tends to be more diverse in its topics.

The procedures to prevent misinformation lie in disaster planning and management [9], a well-established law and regulation system [10], and a proper outlet with sufficient and necessary information. According to scholars, such as Quarantelli, E. L. [11], disaster planning is essential for crisis management, and information flow for communication purposes plays an important role, where “information flow from organizations to the general public” about disasters is usually processed poorly. A lack of organizational consensus among communities and organizations has caused a lot of problems. A consensus must be formed for co-ordination to cope with emergency situations. During an emergency, the need for information grows with public concerns; therefore, sufficient information provided by a proper outlet would greatly ease uncertainty. Moreover, utilizing the information citizens provide would even help in the discovery of important strategies for mitigating the crisis [4]. Similarly, establishing proper information processing mechanisms during crises could help gather community intelligence for problem solving [8].

As an elusive concept, misinformation is generally considered as false information that lacks clear evidence [12]. Although some scholars [13] have argued for further distinguishing disinformation, i.e., false information intentionally fabricated for deception, from misinformation, i.e., unwittingly disseminated false information, the term “misinformation” has been commonly used to represent any false information deliberately or accidentally shared [14], unless the intention to mislead (e.g., meddling with elections) is the focal research point. Under this big umbrella of “misinformation” are many forms of false information with varying degrees of falsehood: gossip, rumor, fake news, hoaxes, legends, myths, and so on [15]. Misinformation is also nuanced in terms of its implied consequences: dread misinformation contains undesirable outcomes such as death, economic collapse, or disasters, whereas wishful misinformation, such as finding a cure for cancer, promises positive results [15,16,17].

As millions of years of evolution have hardwired our nature to be risk aversive, the imminent threats users perceive from dread misinformation will invoke negative emotions and motivate avoidance behaviors to prevent the dire consequences [18], which will in turn impact how users evaluate and react to misinformation from social media outlets [19]. In contrast, the positive emotions induced by wishful misinformation will likely elicit motivation that encourages users to pursue the desired outcomes promised in wishful misinformation [20], including evaluating the wishful misinformation favorably and taking actions to share it.

To generate comprehensible explanations, common methods used for the analysis of text-based data are sentiment analysis and LDA (Latent Dirichlet Allocation) topic modeling, which previous studies on COVID-19 have also applied [21,22,23,24]. For example, some scholars work on sentiment analysis for news during epidemics [22]. Some scholars have analyzed the social media platform Twitter using a sentiment analysis [23,24]. In terms of topic modeling approaches, LDA and STM are widely used in text-based data analysis. Furthermore, there are researchers using LDA to identify major topics on the Twitter platform about COVID-19, others also focus on users’ feedback on online forums, and LDA is used with manual annotation and sentiment analysis to further extract meaningful topic dynamics and sentiment trends [25,26].

With the above previous studies’ analyses, we try to answer the following two research questions in this paper:

RQ1. Comparing major themes and topics of content between online misinformation and newspaper, what are the differences and similarities during the early period of the COVID-19 outbreak?

RQ2. What emotions are revealed from the misinformation on the epidemic outbreak?

## 3. Methods

### 3.1. Subsection Data Collections

News reports. We used the Huike (Wisers) database to collect COVID-19-related reports from news outlets in China, which is one of the largest professional news data service providers for major Chinese news agencies, such as People’s Daily, with an up-to-date collection of more than 1500 different press archives (http://wisenews.wisers.net.cn/, (accessed on 21 February 2020)). Using the search keyword “coronavirus” in title and first paragraph of news content, we gathered a total number of 7791 different newspaper articles from major Chinese news agencies between 1 January 2020 to 20 February 2020.

Social media misinformation data. We also collected 372 pieces of misinformation that emerged from 1 January 2020 to 20 February 2020 from two online platforms: news.qq.com, (accessed on 21 February 2020) and www.dxy.cn, (accessed on 21 February 2020). This is mainly because the former is operated by the most popular social media company in China, Tencent, which closely monitored and reported online misinformation about coronavirus at its early stage, and the latter is a professional health information portal that actively aggregated the latest coronavirus-related misinformation on a daily basis since it broke out in China.

### 3.2. Data Pre-Processing and Preparation

To prepare the data collected for LDA, we applied a few techniques to pre-process the datasets [27,28,29]. As shown in Figure 1, we first conducted word extraction and segmentation with Jieba in Pyhton, Jieba was widely used in many Chinese text mining and sentiment analysis studies. In the second step, we cleaned null content such as missing data from the datasets. In the third step, we removed common stop words such as “is”, “are”, and “them”, from Chinese vocabulary, which have no specific meanings. Lastly, we conducted TF-IDF (term frequency inverse document frequency) transformation to give more weight to those words appearing with high frequencies only in our datasets. TF-IDF is a commonly used pre-processing technique for text mining [30].

### 3.3. Data Analyses

To answer RQ1 and generate human-readable explanations regarding the common themes of the news reports and misinformation about coronavirus in the early stage of the COVID-19 pandemic, we applied the LDA (Latent Dirichlet allocation) topic-modelling analysis to both the news-reports and misinformation datasets.

LDA is a widely used technique in text mining for analyzing data [26,31,32]. One critical question in LDA is to decide the optimal number of topics emerging from a given dataset. While scholars in the past have relied on various creative approaches to this question [33], one commonly adopted method to determine the optimal topical number is to obtain the coherence score, as recommend by Stevens and his colleagues (2012) [34], through calculating the most representative words’ semantic similarities in a topic. The arithmetic mean of these similarities is then calculated to represent the level of consistency for each topic [35].

To answer RQ2, we implemented sentiment analysis to answer RQ2 about the possible emotions that drove the transmission of the misinformation on social media, as emotion has been identified as a notable predictor of the spread of misinformation. The sentiments that emerged from the misinformation data could provide a gateway to understanding the public emotional states expressed in the spread of misinformation on the coronavirus during the early stage of the COVID-19 outbreak in China. Researchers have conducted studies on different emotions expressed in misinformation with a popular lexicon provided by National Research Council Canada (NRC) [27]. This lexicon has been widely used for analyzing emotions expressed in tweets [27], news [28], and customer reviews [29]. We adopted this leading lexicon to analyze eight emotions based on Plutchik’s emotion classification (2001)—anger, fear, anticipation, trust, surprise, sadness, joy, and disgust [30]—to process the misinformation data. The lexicon can process text in different languages including Chinese for the eight types of emotions.

## 4. Results

With the genism package in python, we calculated the coherence value of different topic number [32,33]. When there were eight topics, we found that the coherence value was 0.5109 (see Figure 1); increasing number of topics will result in a growing coherence value, but it increases more slowly after reaching eight as the topic number.

Taking only machine learning into account, the possible results could be without interpretations [33]. By choosing an optimal topic number with both machine learning and interpretability into account [33,34,35], we decided to choose eight as the topic number for our study. After the LDA topic modeling on misinformation and news data, we generated eight topics for each source and further classified them into fewer themes and categories to analyze manually.

For the news data LDA analysis, we found eight topics and grouped them into four categories of themes (see Figure 2 and Table 1). The major theme, accounting for up to 39.40%, is “Prevention and control work, economic influences and society support”, followed by “Prevention and control, government work and notice” (28.80%) as the second biggest theme. The third largest theme was “Prevention and control work, instructions and call for determination” (18.30%) and the fourth was “Cases and development” with a total of 13.50% (see Table 1).

To compare the news data with the misinformation data, we also conducted an LDA analysis to illuminate the major topics and themes. There are eight topics of misinformation that we generated, and we categorized them into three different themes (see Figure 3 and Table 2). The theme with the largest proportion in relation to the COVID-19 outbreak from news media is “Theme 1: virus spread, prevention and control” consisting of 37.7% of all news, “Theme 2: Medical supplies, prevention and control measures” (37.3%) is the second biggest theme, which is followed by the third one “Theme 3: Medical cure, medicine and prevention” (24.9%) (see Table 2).

After the data-cleaning process, we calculated the number of counts of different emotions, the sum number of different emotions, and used matplotlib package (https://matplotlib.org/, accessed on 20 March 2020) in python to visualize the result shown in Figure 4, Figure 5 and Figure 6 for further analysis.

## 5. Discussion

To answer RQ1 and generate a humanly comprehensible explanation, we compared the news data with misinformation data and analyzed citizens’ reception of and search for information during COVID-19’s early period, among which the following caught our attention.

There are some similarities between the two sources of information. Both news media and online misinformation showed concerns for the prevention and control of the virus. For instance, the news dataset’s Theme 1, Theme 2, and Theme 3 all similarly concern the “Prevention and control” category, while the misinformation data show a similar concern: Theme 1 and Theme 2. This illustrates that the press media is posting information with people’s needs and wants in a general direction, trying to build up a robust, harmonious, and healthy environment.

Although from the major themes’ categories, news media and rumors deliver information with greater similarity rather than differentiation, but they emphasize different details. After carefully comparing these “prevention and control” data, we observed quite obvious distinctions in the details.

Regarding misinformation Theme 1, it considers the spread of the virus and the epidemic itself to form prevention and control messages, such as the origin and spread, and many detailed uses of prevention are mentioned, such as avoiding seafood, using alcohol, drinking white wine, Shuang Huanglian (note: one kind of common Chinese medicine for colds), etc. Whereas for Theme 2, the major emphasis lies in medical supplies and transportation controls, such as masks and road controls to stop the transmission of the virus. However, very few news reports concern these personal choices of diet or medicine for the prevention and detection of the virus infection.

The third biggest group of themes for misinformation is “Medical cure, medicine and prevention”, which is mainly about personal choices of medicine, curing methods, and related health information. A great deal of misinformation is about specific medicine for curing the virus, effective prevention methods, and medical choices, some of which even mention being verified by noted experts, such as Dr. Nanshan Zhong. However, in the news-based information, although prevention and control suggestions are given, they are only provided as general instructions without specific advice, especially about personal choices of medicine or medical treatment.

The above two major differences demonstrate a lack of information for normal citizens to obtain enough detailed information towards personalized suggestions on medical decisions during the early period of the COVID-19 outbreak, such as prevention, detection, choice of medicine, and medical cure measures.

Comparing the news and misinformation data, when sufficient information is provided in a newspaper, the less likely it is that misinformation will spread on the same topic, and vice versa. As the different details of the prevention and control topic shows, because of the lack of personal guidance of medical choices in press media, a great deal of misinformation on these topics is spread. Some other emphases of the news articles show the opposite pattern: in areas where newspapers generate a number of reports, misinformation would have less of a potential to increase.

For news data, regarding Theme 2, news reports emphasize government work conducted towards prevention and control. Since there is a sufficient amount of information about this category of news, very little misinformation was spread on this topic.

Likewise, from the news data, regarding Theme 1, newspapers report prevention and control work from a viewpoint of economic and societal influences, such as enterprise influences, society donations, and support. With a relatively large proportion of information for this category of news, misinformation of this kind has less room to increase.

To understand the emotions that netizens revealed when uncertain information was provided and to discuss RQ2, we implemented a sentiment analysis to seek the distribution of different emotions as shown in Figure 7, and to try to analyze what drove the transmission of misinformation on social media. We found that fear was the most dominant emotion, comprising 21.2% of the total misinformation-related emotions. Whereas trust was represented 15.4% in total, which showed people’s concern towards uncertainties, and a similar proportion of 15.2% was represented by the emotion of disgust. Thus, these are the major driving forces for the transmission of misinformation.

However, in news reports, besides actions, the media tried to build up a positive attitude and optimistic belief. For Theme 3, the press media coverage included epidemic prevention and control with a very optimistic standpoint, demonstrating the labor achieved and the progresses made from the community to hospital to government organizations. This kind of news also showed a strong belief and determination to win this battle. Compared to misinformation, these details seldom appear in the transmission. However, there are also forms of misinformation hoping for the best, or fearing for the worst, appearing in the early period of the epidemic outbreak. Through an emotion analysis, we did observe interesting points concerning the seeking and processing of misinformation.

## 6. Conclusions

To understand individuals’ patterns of health communication with online and press media sources, we collected and compared the news data with misinformation data, and analyzed citizens’ information-seeking and -receiving behaviors during the early period of COVID-19, from 1 January 2020 to 20 February 2020. We also conducted an LDA topic modeling analysis and sentiment analysis and provides possible explanations for the driving forces for misinformation and the associated emotions people experienced during the epidemic crisis. Future studies could be conducted to focus on aspects of Media literacy, and to analyze audiences and their responses towards information.

## Figures and Tables

**Figure 1 ijerph-19-09800-f001:**
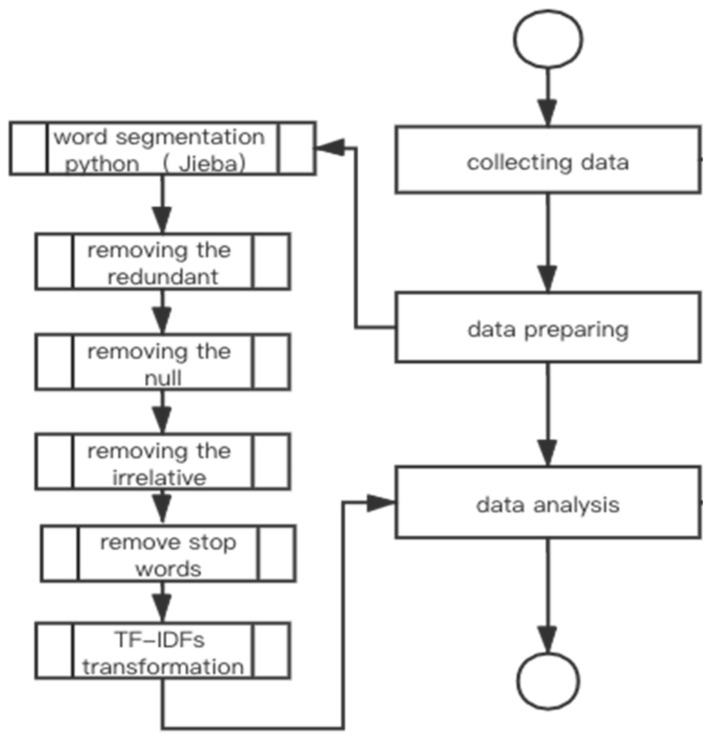
Data Processing Workflow.

**Figure 2 ijerph-19-09800-f002:**
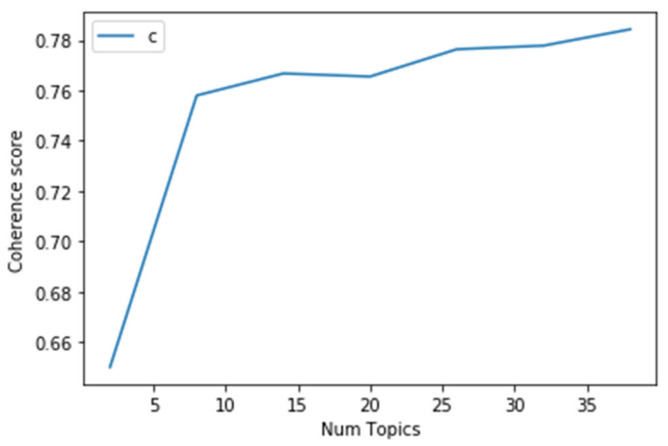
Coherence score for number of topics for news data.

**Figure 3 ijerph-19-09800-f003:**
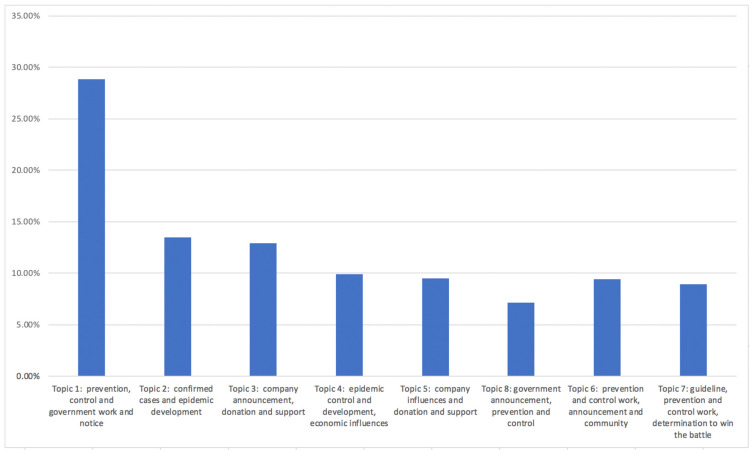
News data LDA analysis topic result.

**Figure 4 ijerph-19-09800-f004:**
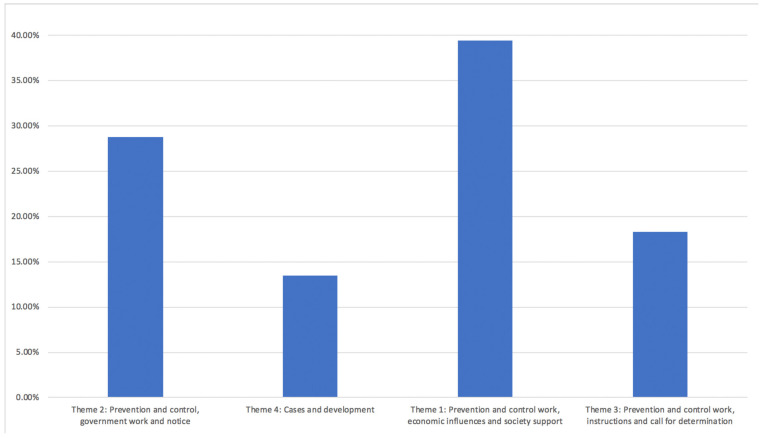
News data LDA analysis theme result.

**Figure 5 ijerph-19-09800-f005:**
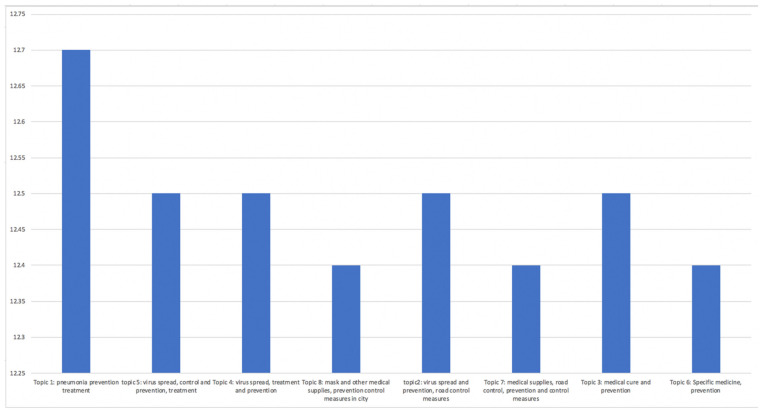
Misinformation data LDA analysis topic result.

**Figure 6 ijerph-19-09800-f006:**
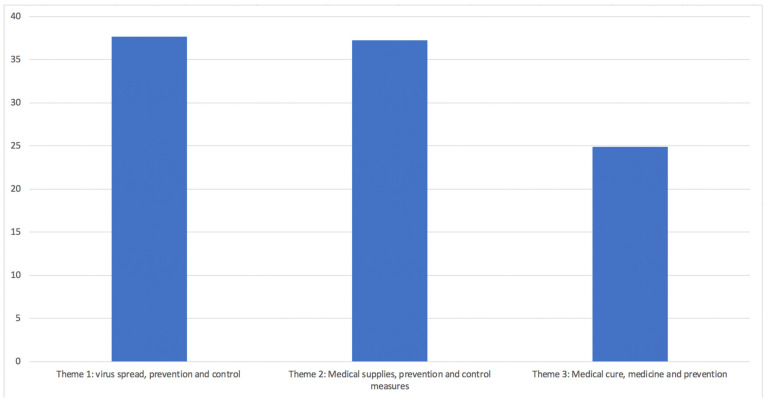
Misinformation data LDA analysis theme result.

**Figure 7 ijerph-19-09800-f007:**
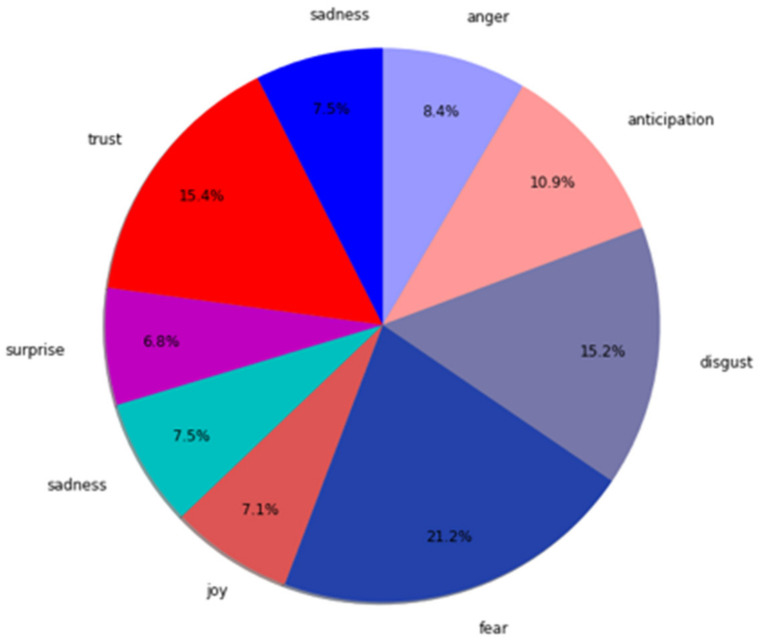
Sentiment proportion of COVID-19 misinformation.

**Table 1 ijerph-19-09800-t001:** News data LDA analysis from 1 January 2020 to 20 February 2020.

Themes	Topic Names	Key Words	Topic Proportion (%)	Theme Proportion (%)
Theme 2: Prevention and control, government work and notice	Topic 1: prevention, control, and government work and notice	Infection, Pneumonia, Coronavirus, Work, Prevention and Control, Command, held meeting, Case, confirmed, outbreak, notice, response, leadership group, Cumulative, My province, News, Press conference	28.80	28.80
Theme 4: Cases and development	Topic 2: confirmed cases and epidemic development	Patient, Cure, my city, Confirmed, Infection, Case, pneumonia, coronavirus, first case, knowledge, cumulative, control and prevention, outbreak, first batch	13.50	13.50
Theme 1: Prevention and control work, economic influences, and society support	Topic 3: company announcement, donation, and support	Announcement, Donation, Support, Resistance, Outbreak, Coronavirus, New type, Pneumonia, Limited, Group shares, Prevention and control, Detection, Response, nucleic acid, technology shares	12.90	39.40
	Topic 4: epidemic control and development, economic influences	Public, Coronavirus, Infection, control, plan, Protection, Pneumonia, Outbreak, Notification, enterprise, return to work, My province, Illegal crimes, Diagnosis, and treatment	9.90	
	Topic 5: company influences and donation and support	Shared company, operation, announcement, affect, coronavirus, outbreak, production, prevention, pneumonia, infection, control, drug, treatment, control	9.50	
	Topic 8: government announcement, prevention, and control	Municipal Committee, Coronavirus, Infection, Outbreak, Announcement, Supervision, Pneumonia, central government, risk, control, return journey, Li Keqiang, China, response, guideline, Joint Control, Joint Defense, close contact, Jiangxi	7.10	
Theme 3: Prevention and control work, instructions, and call for determination	Topic 6: prevention and control work, announcement, and community	prevention and control, outbreak, decision, work, do well, coronavirus, pneumonia, Infection, detection, Home, Community, Notification, Hebei Province, Video, Research, Report, Consultation	9.40	18.30
	Topic 7: guideline, prevention and control work, determination to win the battle	Guideline, Prevention and Control, work, Outbreak, Coronavirus, Infection, according to law, pneumonia, development, manual, medical institution, proposal letter, determination, Resolutely, confront, battle	8.90	

**Table 2 ijerph-19-09800-t002:** Misinformation data LDA analysis from 1 January 2020 to 20 February 2020.

Theme	Topic Names	Key Words	Topic Proportion (%)	Theme Proportion (%)
Theme 1: virus spread, prevention and control	Topic 1: pneumonia prevention treatment	Pneumonia, coronavirus, treatment, infectious, boiling water, resistance, pet, Standard, Public, Smoking, Reservation, Harvard, Rent-free, Fish pond	12.7	37.7
	Topic 5: virus spread, control and prevention, treatment	Prevention, Coronavirus, Doctor, Nanjing, Death, ID card, Bus line, Fuzhou, medical Material, Main road, Seafood, Alcohol, White wine, Infection, High temperature, Humidifier, Shuang Huang Lian, Immunity	12.5	
	Topic 4: virus spread, treatment and prevention	New type Coronavirus, Aerosol, Prevention and control, Academician, Large area, Xiwei, Died, Feng you jing, Body temperature, Airplane, Bee venom, Atomization, Sweater, Playing with snow, Carambola	12.5	
Theme 2: Medical supplies, prevention and control measures	Topic 8: mask and other medical supplies, prevention control measures in city	Mask, Go out, High Way, School, epidemic prevention, closed, public toilet, plague, whole city, Supplier, Nanjing, Home, Huaxi, Block, Guangzhou, Leave, Exposure, Decision, Zhong Nanshan	12.4	37.3
	Topic 2: virus spread and prevention, road control measures	Wuhan, Spread, Patient, Shanghai, Confirmed, seal the city, Hubei, Isolation, Patient, Kill, Protection, Shenzhen, Diaper, Seafood, Detection, Soap, Transportation, High Way, Delay, restrictions	12.5	
	Topic 7: medical supplies, road control, prevention and control measures	Disinfection, Alcohol, Beijing, Spray, Medication, Garlic, Red Cross, United States, Japan, Closed road, Flu, Risk, Supermarket, Antivirus, Vaccine, Guideline, Sanitary napkin, Cold, Li Wenliang, enterprise	12.4	
Theme 3: Medical cures, medicine, and prevention	Topic 3: medical cure and prevention	Outbreak, Hospital, Cure, Virus infection, Libawei, Chlorine dioxide, Antivirus, Zhong Nanshan, China, SARS, Air conditioner, Express parcel, Assistance, Hangzhou, Video, Professor, Shandong, People, Home, Cash, Blood	12.5	24.9
	Topic 6: Specific medicine, prevention	Specific medicine, Virus Infection, Chengdu, Zhong Nanshan, Case, Bus Discontinued, News, Thailand, Mask, Outbreak, Fireworks, Sick, Napkin	12.4

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
