# Peer review of "Health as Battlefield: News and Misinformation in the Early Stage of COVID-19 Outbreak"

_ijerph, 2022, doi:10.3390/ijerph19169800_

Round 1

Reviewer 1 Report

The paper deals with a super interesting and contemporary topic. Well done to the authors for collecting and managing such a large amount of data, as well as for analyzing. 

I really enjoyed reading the paper!

I would suggest consider separating the part of the discussion and the conclusions, or perhaps add a part that focuses only on the conclusions, separately.     

Author Response

Dear Reviewer

Thank you very much for your constructive suggestions and time. We have revised the paper and would like to re-submit it for your consideration. We have carefully read all the comments and addressed all issues as follows. All the changes made within the manuscript are in blue color and reviewer’s words are in red color.

Thanks again for your valuable advice.  

Yours sincerely,

author

Reviewer1: Comments and Suggestions for Authors

The paper deals with a super interesting and contemporary topic. Well done to the authors for collecting and managing such a large amount of data, as well as for analyzing. 

I really enjoyed reading the paper!

I would suggest consider separating the part of the discussion and the conclusions, or perhaps add a part that focuses only on the conclusions, separately.     

Response: 

We agree with the reviewer and add a conclusion at the end as follows:

“6. Conclusion

To understand individuals’ patterns of health communication with online and press media sources, we collected, compared the news data with misinformation data and analyzed citizens’ information seeking and receiving during COVID – 19 early period, from January 1, 2020 to February 20, 2020, we conducted LDA topic modeling analysis and sentiment analysis and provide possible explanations on the driving forces for misinformation and the associated emotions people experienced during the epidemic crisis.”

(line 263 - line 269)

Reviewer 2 Report

The paper tried to do a topic modelling and sentiment analysis of misinformation as well as a topic modelling of traditional news media on COVID-19. However, it has many problems that should be addressed.

Rumors do not equal to misinformation. Rumors are unverified claims, they could be true or false after verification. You cannot use “rumor” and “misinformation” interchangeably.

The authors did not explain why they are comparing traditional news with online misinformation. Do they imply traditional news also contain misinformation or is there an agenda setting between the two? A more logical way to frame the study is that traditional news did not report certain issues, which led to a raised uncertainty towards the issue, which led to rumor.

The authors lack comprehensive literature review on misinformation and therefore the study did not have much theoretical contribution.

The sentiment analysis also did not yield much useful information. What have we learnt from the sentiment analysis? Learning about netizens’ sentiment towards misinformation could help us do what?

The words in Figure 5 are too small to be seen.

Author Response

Dear Reviewer

Thank you very much for your constructive suggestions and time. We have revised the paper and would like to re-submit it for your consideration. We have carefully read all the comments and addressed all issues as follows. All the changes made within the manuscript are in blue color and reviewer’s words are in red color.

Thanks again for your valuable advice.  

Yours sincerely,

author

Reviewer2:Comments and Suggestions for Authors

The paper tried to do a topic modelling and sentiment analysis of misinformation as well as a topic modelling of traditional news media on COVID-19. However, it has many problems that should be addressed.

Rumors do not equal to misinformation. Rumors are unverified claims, they could be true or false after verification. You cannot use “rumor” and “misinformation” interchangeably.

 Response: 

We agree with the reviewer and carefully change or delete rumors related description and add misinformation, for example as follows:

“Misinformation during crisis form and develop for many reasons in virous platforms. Some scholars studied misinformation in crisis during wartime [1], others focused on pandemic, natural disaster [2] or terrorist attacks [3]. Scholars found that online sites as well as mainstream newspaper circulated a number of misleading information for 9-11 crisis [3]. Social media also serve as an arena for rumor spreading, such as twitter [4][5], where social media accelerate the communication process of crisis response and become part of it [4]. It is also detected that the dissemination of rumor on twitter platform are different from news in the same community, in ways that people questioned about misinformation more on social media comparing to news [5].

  1. Related Works

For decades, researchers are looking for features of misinformation spreading online, so that best practices could be taken to fight both the virus and misinformation. Some scholars worked on social media messages on rumor-affirming or rumor-correcting speed to learn the spread model of fake news online ([1]. Others discovered factors causing social media misinformation are information with no clear source, stories with a lot of personal involvement and anxiety [2].“

Procedures to prevent misinformation lies in disaster planning and management ahead[3], well-established law and regulation system[4] as well as proper outlet of sufficient information people need.

(line 25-55)

“Social media misinformation data. We also collected 372 pieces of misinformation emerged from Jan 1, 2020 to Feb 20, 2020 from two online platforms: news.qq.com/ and www.dxy.cn. This is mainly because the former is operated by the most popular social media company TECENT in China that closely monitored and reported online misinformation about coronavirus at its early stage and the latter is a professional health information portal that actively aggregated the latest coronavirus related misinformation on a daily basis when it broke out in China.”

(line 111 - 117)

The authors did not explain why they are comparing traditional news with online misinformation. Do they imply traditional news also contain misinformation or is there an agenda setting between the two? A more logical way to frame the study is that traditional news did not report certain issues, which led to a raised uncertainty towards the issue, which led to rumor.

  Response: 

We agree with the reviewer and carefully added the reason for comparing the traditional news with online misinformation as follows:

“The reason why we compare the traditional news with the misinformation online is that due to the structural differences between the two (e.g., readership, ownership, etc.), traditional news media can only cover certain issues related to the epidemic but not others (e.g., the lab-leaking theory) while online misinformation tend to be more diverse in its topics.”

(line 48 - 52)

The authors lack comprehensive literature review on misinformation and therefore the study did not have much theoretical contribution.

   Response: 

We thank the reviewer for the constructive advice, with the reviewer’s suggestions, we carefully added the following literature review:

As an elusive concept, misinformation is generally considered as false information that lacks clear evidence. [5] Although some scholars [6] have argued to further distinguish disinformation, i.e., false information intentionally fabricated for deception, from misinformation, i.e., unwittingly disseminated false information, the term “misinformation” has been commonly used to represent any false information deliberately or accidentally shared,[7] unless the intention to mislead (e.g., meddling with elections) is the focal research point. Under this big umbrella of “misinformation” are many forms of false information with varying degrees of falsehood: gossip, rumor, fake news, hoax, legend, myth, and so on. [61] Misinformation is also nuanced in terms of its implied consequences: dread misinformation contains undesirable outcomes such as death, economic collapse, or disasters, whereas wish misinformation like finding a cure for cancer promises positive results.[8][9][10].

As millions of years of evolution have hardwired our nature to be risk aversive, [62] the imminent threats users perceive from dread misinformation will invoke negative emotions and avoidance motivation to prevent the dire consequences,[11] which will in turn impact how users evaluate and react to misinformation on social media. [12] In contrast, the positive emotions induced by wish misinformation will likely elicit approach motivation that encourages users to pursue the desired outcomes promised in wish misinformation, [13] including evaluating the wish misinformation favorably and taking actions to share it.

  1. Zeng, L.; Starbird, K.; Spiro, E.S. Rumors at the Speed of Light? Modeling the Rate of Rumor Transmission during Crisis. In Proceedings of the 2016 49th Hawaii International Conference on System Sciences (HICSS); IEEE, 2016; pp. 1969–1978.
  2. Oh, O.; Agrawal, M.; Rao, H.R. Community Intelligence and Social Media Services: A Rumor Theoretic Analysis of Tweets during Social Crises. MIS quarterly 2013, 407–426.
  3. Jin, Y.; Liu, B.F. The Blog-Mediated Crisis Communication Model: Recommendations for Responding to Influential External Blogs. Journal of public relations research 2010, 22, 429–455.
  4. Liang, J.; Yang, M. On Spreading and Controlling of Online Rumors in We-Media Era. Asian Culture and History 2015, 7, 42.
  5. Nyhan, B.; Reifler, J. When Corrections Fail: The Persistence of Political Misperceptions. Polit Behav 2010, 32, 303–330, doi:10.1007/s11109-010-9112-2.
  6. Hjorth, F.; Adler-Nissen, R. Ideological Asymmetry in the Reach of Pro-Russian Digital Disinformation to United States Audiences. Journal of Communication 2019, 69, 168–192, doi:10.1093/joc/jqz006.
  7. Southwell, B.G.; Thorson, E.A.; Sheble, L. Misinformation and Mass Audiences; University of Texas Press, 2018; ISBN 978-1-4773-1456-2.
  8. DiFonzo, N.; Robinson, N.M.; Suls, J.M.; Rini, C. Rumors About Cancer: Content, Sources, Coping, Transmission, and Belief. Journal of Health Communication 2012, 17, 1099–1115, doi:10.1080/10810730.2012.665417.
  9. Allport, G.W.; Postman, L. The Psychology of Rumor; The psychology of rumor; Henry Holt: Oxford, England, 1947; pp. xiv, 247;.
  10. Knapp, R.H. A Psychology of Rumor. The Public Opinion Quarterly 1944, 8, 22–37.
  11. Scholer, A.A.; Cornwell, J.F.M.; Higgins, E.T. Should We Approach Approach and Avoid Avoidance? An Inquiry from Different Levels. Psychological Inquiry 2019, 30, 111–124, doi:10.1080/1047840X.2019.1643667.
  12. Bordia, P.; DiFonzo, N. When Social Psychology Became Less Social: Prasad and the History of Rumor Research. Asian Journal of Social Psychology 2002, 5, 49–61, doi:10.1111/1467-839X.00093.
  13. Maio, G.R.; Esses, V.M. The Need for Affect: Individual Differences in the Motivation to Approach or Avoid Emotions. Journal of Personality 2001, 69, 583–614, doi:10.1111/1467-6494.694156.

(page1 line 66 - 85)

The sentiment analysis also did not yield much useful information. What have we learnt from the sentiment analysis? Learning about netizens’ sentiment towards misinformation could help us do what?

   Response: 

We thank the reviewer for the constructive suggestion, we carefully added discussion on netizens’ sentiment analysis as follows:

“To understand emotions netizens revealed when uncertain information is provided and to discuss RQ2, we implemented sentiment analysis to seek distribution of different emotions and try to analyze what drove the transmission of the misinformation on social media. We found fear is the most dominant emotion taking 21.2% of the total misinformation emotion. While, trust emotion is revealed with 15.4% in total which showed people’s concern towards uncertainties, similar proportion of 15.2% is demonstrated with disgust emotion. And these are the major driving forces for the transmission of misinformation.

However, in news report, besides actions, the media is trying to build up a positive attitude and optimistic belief. With Theme 3: press media coverage includes epidemic prevention and control works with a very optimistic stand point, demonstrate works done and progresses made from community, hospital to government organizations. These kinds of news also showed strong belief and determination to win this battle. Comparing to misinformation, these details seldom appear in the transmission. However, there are misinformation hoping for the best, or fearing for the worst appearing in the early period of the epidemic outbreak. Through emotion analysis, we do observe interesting points on understanding of the seeking and processing of misinformation.”

(line 246 - 262)

The words in Figure 5 are too small to be seen.

 Response: 

We thank the reviewer for pointing this out, we carefully changed the Figure 5 file to make the content clear.

Figure 5. Misinformation data LDA analysis topic result.

(please refer to the word document attached for the figure 5)

Reviewer 3 Report

Dear colleagues,

Media literacy is missing here: "Procedures to prevent rumors lies in disaster planning and management ahead[8], 49 well-established law and regulation system[9] as well as proper outlet of sufficient information people need." (pg.2/11) Do you really think that media illiterate person will judge, in proper way, certain information, if you: 1. repeat; 2. fill it up with facts and figures; 3. and by proper means conducted? Without media literacy, the information will bounce out from their thoughts as a soccer ball, far away from them. Critical thinking has to exist which will judge and make his/her own decision, after receiving information. Not to have information per se. It is like sawing the best seed on the dust of Sahara desert under 50 degrees Celsius. What do you think, will it survive? If you use, media literacy within your research questioning and quantitative analyses, you would have amazing results about how people (in general, not just in China) are easy to be manipulated because of their basic ignorance and not knowing how manipulation works. After you do that, your research would be more serious and more fullfilled with proper information, You started with assumption that all people are media literate (my conclusion, because I do not see anywhere source for media literacy within your text-paper) and, eo ipso, conclusion is wrong, without analysing issue of the media literacy. What causes missinformation? Is it: a) capability of the person/company who places it or b) lack of understanding by media illiterate "consumers" of the information? Answer is "b". Why? Because, media literate person will know how to prevent and hot to conclude against missinformation (your writings of pg. 7/11). You cannot defend that you have done just a research to find out the reflections on the issue. You have to have possibility to analyse media literacy in regard the informational issues and include that within the research. Maybe you have it, but I cannot see that within your paper. Include it, please. p.s. All other focus of yours was more than good within the paper. Just include media literacy, please. 

Author Response

Dear Reviewer

Thank you very much for your constructive suggestions and your valuable time. We have revised the paper and would like to re-submit it for your consideration. We have carefully read all the comments and some major changes are made, such as: future study is discussed, literature review is revised, conclusion is separated and added, figures and tables are updated, etc.

All the changes made within the manuscript are in blue color and reviewer’s words are in red color.

Thanks again for your valuable advice.  

Yours sincerely,

author

Reviewer3:Comments and Suggestions for Authors

Dear colleagues,

Media literacy is missing here: "Procedures to prevent rumors lies in disaster planning and management ahead[8], 49 well-established law and regulation system[9] as well as proper outlet of sufficient information people need." (pg.2/11) Do you really think that media illiterate person will judge, in proper way, certain information, if you: 1. repeat; 2. fill it up with facts and figures; 3. and by proper means conducted? Without media literacy, the information will bounce out from their thoughts as a soccer ball, far away from them. Critical thinking has to exist which will judge and make his/her own decision, after receiving information. Not to have information per se. It is like sawing the best seed on the dust of Sahara desert under 50 degrees Celsius. What do you think, will it survive? If you use, media literacy within your research questioning and quantitative analyses, you would have amazing results about how people (in general, not just in China) are easy to be manipulated because of their basic ignorance and not knowing how manipulation works. After you do that, your research would be more serious and more fullfilled with proper information, You started with assumption that all people are media literate (my conclusion, because I do not see anywhere source for media literacy within your text-paper) and, eo ipso, conclusion is wrong, without analysing issue of the media literacy. What causes missinformation? Is it: a) capability of the person/company who places it or b) lack of understanding by media illiterate "consumers" of the information? Answer is "b". Why? Because, media literate person will know how to prevent and hot to conclude against missinformation (your writings of pg. 7/11). You cannot defend that you have done just a research to find out the reflections on the issue. You have to have possibility to analyse media literacy in regard the informational issues and include that within the research. Maybe you have it, but I cannot see that within your paper. Include it, please. p.s. All other focus of yours was more than good within the paper. Just include media literacy, please. 

   Response: 

We sincerely thank the reviewer, as this is a very good suggestion, but our data source platform does not have access to any users’ information now, so we cannot perform this kind of analysis at this stage. But it is a great future direction, that we need to follow up and continue the study on this topic. So, we’ve added the following future plan at the end:

“Future studies could be done to focus on aspects of media literacy, to analyze audience and their responses towards information as well.”

(line269-271)

Round 2

Reviewer 2 Report

This version has been significantly improved.

Reviewer 3 Report

They have replied in a good manner.